# Effects of Physical Activity on Physical and Mental Health of Older Adults Living in Care Settings: A Systematic Review of Meta-Analyses

**DOI:** 10.3390/ijerph20136226

**Published:** 2023-06-26

**Authors:** Nounagnon Frutueux Agbangla, Marie-Philippine Séba, Frédérique Bunlon, Claire Toulotte, Sarah Anne Fraser

**Affiliations:** 1University of Artois, University of Lille, University of Littoral Côte d’Opale, ULR 7369—URePSSS—Unité de Recherche Pluridisciplinaire Sport Santé Société, Liévin, F-62800 Liévin, France; claire.toulotte@univ-artois.fr; 2Institut des Sciences du Sport-Santé de Paris (URP 3625), Université Paris Cité, F-75015 Paris, France; marie-philippine.seba@u-paris.fr; 3Independent Researcher, France; frederiquebunlon@gmail.com; 4Interdisciplinary School of Health Sciences, Faculty of Health Sciences, University of Ottawa, Ottawa, ON K1S 5S9, Canada; sarah.fraser@uottawa.ca

**Keywords:** physical activity, nursing homes, long-term care homes, older adults, umbrella review

## Abstract

Previous studies included in meta-analyses have highlighted the effects of physical activity on the physical and psychological health of older adults living in care settings. We conducted a systematic review of meta-analyses, of institutionalized older adults, to evaluate and conduct a narrative synthesis of the results of these meta-analyses. A literature search was conducted in three databases (PubMed, Web of Science, and Cochrane Library) until 22 March 2023. After screening the identified articles with the PRISMA criteria filters, we included 11 meta-analyses in this systematic review. Higgins’ (2013) assessment tool of the meta-analyses demonstrates that the studies were of good quality although future meta-analyses need to provide more information on the treatment of missing data. A summary of the results of these meta-analyses shows that physical activity reduces the rate of falls, mobility issues, functional dependence, and depression, and improves health status. Future studies need to focus on new ways to promote and adapt physical activities to increase the participation of older adults in care settings.

## 1. Introduction

Aging is heterogeneous, while some older adults will remain socially, physically, and cognitively active [1,2], others may show frailty. Frailty is “a state of increased vulnerability that predisposes the individual to functional decline and ultimately leads to dependence or death” [3]. Complications of a frailty diagnosis may include increased dependence and an inability to stay home alone, which may, over time, lead to older adults being admitted to a nursing home or a long-term care home. According to the National Institute on Aging, “a nursing home is a facility that provides [4] a wide range of health and personal care services. Rehabilitation services, such as physical, occupational, and speech therapy, are also available” [4]. Comparatively, a long-term care home involves a variety of services designed to meet a person’s health or personal care needs during a short or long period of time [5]. In this review, the term care settings (CS) will be used to refer to both types of facilities. A transition from the community into CS is not without consequences as it can lead to a higher risk of falls, fractures, loss of independence, and depressive symptoms. Older adults in CS have a higher risk of falls [6] which are, “an event that results in a person coming to rest inadvertently on the ground or floor or other lower level” [7]. In addition, CS can lead to other problems such as fractures [8], avoidance of daily activities, loss of independence, reduced social activities, and injuries [9,10]. Approximately 30–50% of residents, aged 65 and above, have fallen at least once a year, and 12–40% have experienced recurrent falls in CS [11]. The reported number of residents who fall in CS is three times greater than the number of older adults living in community settings [12]. In addition to poorer physical health outcomes, a high prevalence of depressive symptoms is also observed in residents [13]. In short, many older adults living in CS may encounter physical (falls, fractures, loss of independence) and psychological (depression) issues. To manage these health issues, various non-pharmacological interventions such as physical activity (PA) can be implemented. PA is defined as, “Any body movement generated by the contraction of skeletal muscles that raises energy expenditure above resting metabolic rate. It is characterized by its modality, frequency, intensity, duration, and context of practice” [14,15]. Several studies carried out on older adults in CS have supported the benefits of PA on mental health [16] and physical health [17,18]. According to these studies, PA not only improved overall cognitive functioning but also reduced depression and anxiety [16]. 

Even though studies have demonstrated the benefits of PA in older adults in CS, meta-analyses are needed to obtain an accurate estimate of the effects of the intervention based on the effect sizes of the included studies [19]. Meta-analyses provide more reliable results than single studies since they are based on a set of similar studies, which allows not only a large sample and a shorter confidence interval but also high precision [20]. Therefore, several meta-analyses have been carried out in the literature to ensure the accuracy of the effect of activity on physical (balance, strength, mobility, endurance) and psychological (depression) parameters [10,21,22,23,24,25,26]. However, included studies must be methodologically rigorous for the results of meta-analyses to be valid. According to Crombie and Davies, a good meta-analysis should comprehensively identify relevant studies, look for the presence of heterogeneity, and explore the robustness of the results using sensitivity analysis [19]. Although there are several meta-analyses in the literature that have tested the effect of PA on older adults in CS, few systematic reviews have qualitatively evaluated these meta-analyses. Such an evaluation seems necessary since the results of meta-analyses are often used to make recommendations for practicing PA. This systematic review, therefore, had two main objectives: (1) the first aim was to qualitatively assess the meta-analyses conducted with older adults living in CS using Higgins’ qualitative assessment tool [27]; (2) the second aim was to conduct a narrative synthesis of the main results of the meta-analyses to identify gaps in the literature that should be addressed in future research.

## 2. Materials and Methods

### 2.1. Search Strategy

Two authors (N.F.A. and F.B.) searched for the meta-analyses in the PubMed, Web of Science, and Cochrane Library databases. The last search was performed on 22 March 2023 and considered all articles before that date. The following were the keyword matches during the search process: “Physical activity” AND “Nursing homes” AND “older adults”; “Physical exercise” AND “long-term care facilities” AND “older adults”; “Physical activity” AND “long-term care facilities” AND “older adults”; “Physical exercise” AND “Nursing homes” AND “older adults”; “Physical activity” AND “Nursing homes” AND “elderly”; “Physical exercise” AND “long-term care facilities” AND “elderly”; “Physical activity” AND “long-term care facilities” AND “elderly”; and “Physical exercise” AND “Nursing homes” AND “elderly”. Moreover, the results were filtered to include studies with “Meta-analysis” and “65+ Aged”. The search strategy retrieved 106 meta-analyses on PubMed, 466 meta-analyses on the Web of Science, and 9 meta-analyses on Cochrane Library (see Appendix A).

### 2.2. Study Selection and Eligibility Criteria

After identifying the meta-analyses (582 items), we proceeded to the selection. The selection involved screening titles and abstracts to remove duplicates (317 items) and irrelevant (222 items) meta-analyses. Irrelevant meta-analyses were either systematic reviews or narrative reviews or meta-analyses that focused on wrong outcomes or wrong populations. The studies that were not duplicates or eliminated by screening (n = 43) were reviewed in full text by two authors (N.F.A. and F.B.) to include meta-analyses that met the following criteria: (a) the average age of the participants must be ≥65 years; (b) the meta-analyses must be conducted on studies that included older adults living in CS; (c) the language of the publication must be English; (d) the meta-analyses must investigate the effect of a PA program on at least one of the two variables of interest: (1) physical outcomes (mobility, balance, gait, strength, fall, fear of falling, and activities of daily living) or (2) mental health outcomes (depression and quality of life); (e) meta-analyses should not include studies with community-dwelling participants. Thus, studies were excluded if participants were in community housing and not only in CS. In all, 11 meta-analytic studies met the different criteria. The selection procedure is illustrated in Figure 1. Finally, the quality of the included meta-analyses was assessed by three authors (M.-P.S., N.F.A., and F.B.) using the Higgin’s tool which is based on questions from the AMSTAR tool and Cochrane Handbook for Systematic Reviews of Interventions [27].

### 2.3. Data Extraction

Three authors (M.-P.S., N.F.A., and F.B.) extracted the data from the selected meta-analyses using a standardized extraction form. The standardized extraction form contained the following items: the article citation, characteristics of the studies included in the meta-analysis, the physical intervention, data sources, analysis of individual studies by the meta-analysts, meta-analysis, reporting and interpretation, and the major outcomes. At the end of the data extraction, extraction by the three authors was compared to ensure accuracy in the extraction of study details across authors. Finally, any disagreements about inclusion or extraction were resolved through discussion.

## 3. Results

### 3.1. Characteristics of the Meta-Analyses

Eleven meta-analyses that met the selection criteria were included in this systematic review. They were conducted by researchers affiliated with institutions in countries such as Austria, Australia, Canada, China, South Korea, Spain, the United States, and the United Kingdom. These meta-analyses analyzed a total of 235 studies conducted in various CS [28,29,30,31,32,33,34,35,36,37,38]. The total number of participants was 56,241 with an average age range between 69 and 90 years. However, if we avoid repeat counting of the participants of studies that are included in several meta-analyses, we are left with 44,477 participants. The meta-analyses tested the effects of several interventions on outcomes of interest that relate to mobility, falls, activities of daily living, health status, quality of life, and depression. Almost all meta-analyses included studies that tested the effects of combined non-pharmacological interventions [28,29,30,31,32,33,34,35,36,37,38]. Interventions ranged between four weeks and 24 months with a frequency of one to seven sessions per week and each session lasted between 15–60 s and 150 min. The characteristics of each meta-analysis have been summarized in Appendix A.

### 3.2. Quality of Meta-Analyses

The reporting of meta-analysis quality in text or table format in this section is based on Higgins’ tool [27] and a recent systematic review of meta-analyses that also used the Higgins tool [39].

#### 3.2.1. Data Sources

The literature search strategy is one of the criteria for assessing the quality of a meta-analysis [19]. The literature search strategy for the 11 meta-analyses considered in this systematic review was based on specific eligibility criteria for participant characteristics, study design, outcomes, and interventions. In addition, these eligibility criteria were supplemented by exclusion criteria. Indeed, 90.9% of meta-analyses included in this systematic review (10/11) used exclusion criteria which were, among others, language, which was limited to English, the publication status (peer-reviewed and published article), and other criteria (chronic-specific disease, other intervention than exercise, no control group, participants living in communities, inadequate information, case reports, and case series studies). Using these criteria, the meta-analysts searched for articles in the published literature, and online repositories, and by contacting authors directly if the article was inaccessible online or through library services. The main databases considered in the meta-analyses were PubMed/Medline, Embase, Central Cochrane Library, and Science Citation Database. Other databases such as Allied and Complementary Medicine Database, Cumulative Index of Nursing and Allied Health Literature, Physiotherapy Evidence Database, Applied Social Sciences Index and Abstracts, International Bibliography of the Social Sciences, PsycINFO, Database of Abstracts of Reviews of Effects, Healthcare Management Information Consortium, NHS Economic Evaluation Database, Health Technology Assessment database, ISI Web of Knowledge, Google Scholar, Index to Theses, ProQuest dissertations and theses, SportDiscuss, Scopus, KoreaMed, KMbase, Korean Studies Information Service System, Korea Institute of Science and Technology Information, SinoMed, China National Knowledge Infrastructure, China Biology Medicine, Chongqing, Occupational therapy seeker, and Wanfang were also consulted during the data research phase. Lastly, article selection and data extraction were performed independently by at least two authors. In short, the literature strategy used in the 11 meta-analyses was adequate, understandable, and minimized or identified biases in study location and assessment. All information regarding the literature search strategy used by the meta-analysts is summarized in Table 1 below.

#### 3.2.2. Analysis of Individual Studies by the Meta-Analyst and General Meta-Analysis

Five out of the 11 meta-analyses (45.45%) considered in this systematic review reported missing data. To address the problem of missing data, the meta-analysts contacted the authors of the selected studies to request the missing data. If they did not receive the missing data, they excluded that study. The remaining meta-analyses (6/11, 54.54%) did not address missing data in their sample. Regarding double counting of included studies in analyses, this was controlled for nine out of 11 meta-analyses (81.81%), not mentioned in one meta-analysis (9.09%), and unclear in one meta-analysis (9.09%). Heterogeneity was assessed in all meta-analyses by the I^2^ statistic which, “describes the percentage of variation across studies that is due to heterogeneity rather than chance” [27]. However, a few meta-analyses supplemented the I^2^ statistic with visualization, i.e., forest plots (1/11, 9.09%), Chi^2^ (2/11, 18.18%), and Cochran’s Q statistic (5/11, 45.45%). A clear strategy to address statistical heterogeneity was described in eight meta-analyses (72.72%) and was unclear in three meta-analyses (27.27%). The synthesis analyses were classic-basic (weighted averages, peto method, Manted-haenszel, weighted linear regression, use of RevMan or metan in Stata) in nine out of 11 meta-analyses (81.81%) and classical-advanced (mixed models, random-effects meta-regression, Hartung-Knapp, Biggerstaff-Tweedie, Hardy-Thompson) in two out of 11 meta-analyses (18.18%). Still based on the Higgins criteria, we noted that subgroup analyses performed well in nine meta-analyses (81.81%) and were not applicable in two meta-analyses (18.18%). Finally, the accuracy of the interpretation of subgroup analyses was correct in nine meta-analyses (81.81%). 

Regarding the type of studies included in the meta-analyses in our systematic review, 100% of the meta-analyses included randomized control trials. We also noted the presence of cluster randomized control trials in five meta-analyses (45.45%), cross-over trials in one meta-analysis (9.09%), and other types of studies (interventional, longitudinal, quasi-experimental design) in six meta-analyses (54.54%). Each of the studies included in the meta-analyses was assessed for risk of bias by the meta-analysts using either a Jaded scale [40], Cochrane risk of bias tool [41], or PEDro scale [42]. These tools (Jaded scale and Cochrane risk of bias tool) were used to evaluate allocation sequence generation, allocation sequence concealment, blinding, attrition, and others. The number of meta-analyses that made these assessments has been specified in Table 2. To test the effects of the different studies, 100% of the meta-analyses used the random-effect model, which is a statistical model used when the authors’ objective is to generalize the results by estimating the mean of the effect size distribution [43,44]. In addition, 27.27% of the meta-analyses also used the fixed-effect model, which tests for a common effect between the studies included in the meta-analysis [43,44]. Finally, 9.09% of the meta-analyses used random-effect meta-regression. All the analyses performed in the different meta-analyses are specified in Table 2. Regarding reporting bias, several tools were used as specified in Table 2. The most frequently used tools were the funnel plots (8/11, 72.72%) followed by the Egger test (5/11, 45.45%). 

Other important indicators, such as issues for continuous data, binary, time-to-event data, ordinal data, and indirect comparisons, were also considered during the evaluation of the meta-analyses. These indicators were identified and are presented in Table 3 based on the meta-analyses and the data used in the meta-analyses.

In short, the assessment performed by the Higgins tool [27] demonstrates that the meta-analyses included in this systematic review are of very good quality since the literature search strategy, the individual analysis of the studies, the method of meta-analysis, and the interpretation of the results were well executed (see Table 4).

### 3.3. Effects of Interventions on Outcomes

When the results of the meta-analyses are examined, findings support the preventative effect of PA on the rate of falls; this preventive effect is greater when the program is combined and includes balance, strength, endurance, stretching, and walking [32,34,38] and is practiced for one to three months or more than six months [34,38]. In addition to the preventive effect, exercise also significantly reduces the number of falls and fallers [28,33,37]. However, this beneficial effect of PA is not observed in older adults with cognitive impairment [33]. While these meta-analyses have supported the beneficial effect of exercise on falls, the meta-analysis by Cao et al., which analyzed studies proposing whole-body vibration, weight training, balance exercises, and Tai Chi exercise as interventions, was unable to demonstrate the effect of exercise on preventing falls [36]. In addition to reducing falls, PA also promotes a significant increase in postural balance, muscular strength of the lower limbs [37], functional mobility [31,35,37], and functional independence [30,31]. As a result, older adults are more independent in their activities of daily living and mobility. Exercise decreases depressive symptoms in older adults with a small effect size [29]. Finally, PA helps to improve health status, as assessed by the number of frail older adults [37]. In the sampled studies, more meta-analyses have tested the effects of exercise on the physical health of older adults in CS than on their mental health. Finally, all the results and the effect sizes of the meta-analyses are summarized in Table 5. 

## 4. Discussion

The goal of this systematic review was to qualitatively assess and summarize the findings of meta-analyses conducted on the effects of PA on the physical and mental health of older adults living in CS. After assessing the literature strategy used in the different meta-analyses, we found that the data were adequately reported and minimized bias in the selection process. However, not all meta-analyses considered the grey literature. Excluding the grey literature could lead to relevant information being omitted. We also noted that while most of the meta-analyses focused on English-only articles, other meta-analyses considered articles written in German, Korean, or Chinese in addition to English [32,33,38]. A more inclusive language criterion allows for a somewhat more generalizable result for a wider population of older adults. When examining the designs of the studies included in the meta-analyses many did not have the same experimental design resulting in some heterogeneity in the studies. However, heterogeneity was controlled in the various meta-analyses. We also observed a low number of studies (<14) in certain meta-analyses [30,34,35,36,37,38], which could be explained by the limited number of studies conducted with older adults living in CS and the feasibility of PA programs in this setting [45]. Regarding the individual analysis of the studies, all the meta-analyses assessed the risk of bias using appropriate tools (Jaded scale and Cochrane risk of bias tool), which allowed the identification and reporting of any study bias. In addition, reporting bias was also assessed using funnel plots, the Egger test, and other methods. Finally, the meta-analyses used appropriate analytical models (fixed-effect model and random-effect model) to test fixed and random effects [43,44]. However, it should be noted that not all meta-analyses specified how missing data were handled and whether the authors avoided double counting. In short, it is important to specify that the few limitations raised in these meta-analyses, concerning heterogeneity and the treatment of missing data, do not detract from the validity of the results of the meta-analyses due to their methodological rigor.

The meta-analyses considered in this systematic review have demonstrated, with few exceptions, that exercise is beneficial to the physical and psychological health of older adults living in CS. In terms of physical health, exercise (aerobic, resistance, flexibility, stretching, mobility, and whole-body vibration), program duration (4–52 weeks), frequency (1–7 sessions/week), and time per session (6–90 min), prevents and reduces falls among this population [28,32,33,34,37,38] and improves functional independence [30,31]. The results of these meta-analyses complement those of the meta-analysis of Sherrington et al., which showed the beneficial effect of PA on falls in community-dwelling older adults [46]. Furthermore, these findings agree with those of Stubbs et al. [47] who also highlighted the positive effects of exercise on falling in an umbrella review of meta-analyses conducted among older adults in CS. The effect of exercise on physical health could be explained by the fact that exercise reduces physical disabilities and functional limitations by strengthening muscles and improving cardiorespiratory fitness, flexibility, and postural balance [34,37]. PA would, therefore, promote compensatory mechanisms against falling [32]. We also found that in addition to programs consisting of resistance, endurance, balance, flexibility, and walking exercises, programs with whole-body vibration exercises would improve functional mobility [35]. According to these authors, the whole-body vibration of the body would trigger the “tonic-vibratory reflex”, which allows the activation of several motor units and, thereby, a large neuromuscular response. One could therefore think that this great neuromuscular response would explain the improvement in functional mobility. Taken together, the meta-analyses included studies that used different physical interventions and reached the same result regarding the effect of PA on the physical health of older adults living in CS. 

Li et al.’s meta-analysis on depression demonstrated that beyond physical health, exercise (i.e., a mix of aerobic, muscle and bone strengthening, stretching/flexibility, Tai Chi, and yoga), program duration (4–64.5 weeks), frequency (1–7 sessions/week), and time per session (20–90 min), improves depressive symptoms in older adults living in CS [29]. However, this effect on depression is small. Several mechanisms could explain this beneficial effect of PA on depression. Neurobiologically, the exercise would increase the volume of the hippocampus, prefrontal cortex, and cingulate cortex involved in the limbic neural network that mediates the effect of exercise on depressive symptoms [29,48,49]. Another mechanism that may explain the effects of exercise on depression is that exercise allows the release of brain-derived neurotrophic factors and other hormones such as dopamine and serotonin, which reduce depressive symptoms [16,50,51,52]. Exercise may also affect anxiety in older adults. However, in the present systematic review, we were unable to identify a meta-analysis on anxiety and exercise that had been conducted with older adults in CS. A recent meta-analysis that included studies with a truly heterogeneous population (community-dwelling and those in CS) showed that PA diminished anxiety symptoms [53]. Although this meta-analysis was not included in our systematic review, it is noteworthy as it included community-dwelling participants and it demonstrates that exercise may affect anxiety in this population. 

The strength of our work is that our systematic review is one of the few examining meta-analyses on older adults in CS. This area needs further development and research to support older adults that live in these settings and have higher care needs. Our review does have some limitations. For example, we conducted our literature search on three databases and included meta-analyses written exclusively in English. The results, analyses, and interpretations of our review are therefore limited to these databases and language choice. In addition, the fact that we limited our included studies to those with older adults in CS meant that we had a limited selection of meta-analyses. Finally, as specified in some meta-analyses, certain biases may lead to an overestimation of positive effects [31].

The main results of our systematic review support that PA is beneficial for the physical and psychological health of older adults in CS. Despite these benefits of exercise, engaging and adapting PA programs for older adults in these settings is a challenge for CS managers. Although the barriers to engagement in PA among older adults are studied and known [54], the problem persists and may be exacerbated in CS. Future studies on PA in older adults in CS should focus on fun physical practices such as games to test not only their attractiveness but also their effect on the physical and psychological health of older adults in CS. While a few studies exist in the literature [55,56], replication studies are needed to support the effects of these practices in older adults who live in CS. It would also be interesting to collect the perspectives of older adults who live in these settings, via an intervention to capture their reasons for engaging in PA. We also found that none of the included meta-analyses analyzed the relationship between physical and mental health. Research supports a strong two-way relationship between mental health (depression and anxiety) and physical health [57]. Future studies with this population of older adults need to focus more on the relationship between physical and mental health. Finally, we found that none of the meta-analyses included in this review presented data according to the physical and/or cognitive abilities or pathologies of older adults in CS. Future literature reviews should address this gap in the literature.

## 5. Conclusions

To conclude, almost all the meta-analyses considered in this systematic review highlight the benefits of PA on the rate of falls, functional independence, depression, and the quality of life of older adults in CS. Future studies must now focus on types of physical activities and strategies to engage older adults living in CS in PA. 

## Figures and Tables

**Figure 1 ijerph-20-06226-f001:**
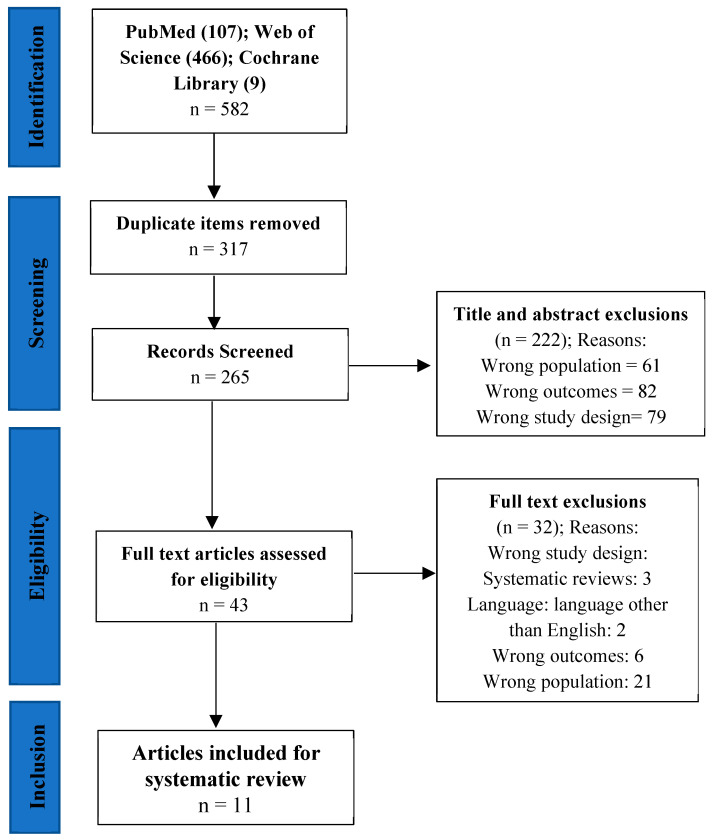
PRISMA diagram of the selection process.

**Table 1 ijerph-20-06226-t001:** Resource search criteria in meta-analyses.

Criteria	Frequency (%)	
	Yes (%)	No (%)
**Eligibility criteria were stated and suitably specific for**		
Participants	11 (100)	-
Experimental intervention (s)	11 (100)	-
Comparator intervention (s)	10 (90.9)	1 (9.09)
Outcomes	10 (90.9)	1 (9.09)
Study designs	11 (100)	-
**Further restrictions on eligibility on studies or reports**		
Specific restriction	-	-
Publication status restriction	4 (36.36)	7 (63.63)
Language restriction	6 (54.54)	5 (45.45)
Other restriction	9 (81.81)	2 (18.18)
**Data for meta-analysis were sought from**		
Published literature	11 (100)	-
Online repositories	10 (90.9)	1 (9.09)
Correspondence with trialists	1 (9.09)	10 (90.9)
In-house IPD ^a^	-	11 (100)
Other IPD	-	11 (100)
**The search for trials included**		
Bibliographic databases	11 (100)	-
Grey literature	2 (18.18)	9 (81.81)
The web	2 (18.18)	9 (81.81)
In-houses collections	-	11 (100)
Reference lists	9 (81.81)	2 (18.18)
Hand searching	8 (72.72)	3 (27.27)
Correspondence with industry	-	11 (100)
Other correspondence	-	11 (100)
Other sources	1 (9.09)	10 (90.9)
**Which bibliographic databases are mentioned**		
PubMed/MEDLINE	11 (100)	-
EMBASE	9 (81.81)	2 (18.18)
CENTRAL/Cochrane Library	10 (90.9)	1 (9.09)
Science Citation Database	3 (27.27)	8 (72.72)
Other	11 (100)	-

IPD ^a^ = Institutional Profiles Database.

**Table 2 ijerph-20-06226-t002:** The methods used in risk of bias and reporting bias.

Criteria	Frequency (%)	
	Yes (%)	No (%)
**Risk of bias (quality assessment) or eligibility criteria include**		
Generation of allocation sequence	10 (90.9)	1 (9.09)
Concealment of allocation sequence	11 (100)	-
Blinding	10 (90.9)	1 (9.09)
Attrition/dropout/ITT ^b^	9 (81.81)	2 (18.18)
Other	7 (63.63)	4 (36.36)
**The synthesis methods used in the paper**		
Pooling (no stratification by study)	-	-
Fixed-effect meta-analysis	3 (27.27)	8 (72.72)
Random-effect meta-analysis	11 (100)	-
Fixed-effect meta regression	-	-
Random-effect meta regression	1 (9.09)	10 (90.9)
**Tools was used for assessed reporting bias**		
Funnel plots	8 (72.72)	3 (27.27)
Egger test	5 (45.45)	6 (54.54)
Begg-mazumdar rank correlation test	1 (9.09)	10 (90.9)
Other Funnel plots asymmetry test	-	-
Trim and Fill	1 (9.09)	10 (90.9)
Other	5 (45.45)	6 (54.54)

ITT ^b^ = intent to treat.

**Table 3 ijerph-20-06226-t003:** Testing and reporting issues based on data type.

Criteria	Yes (%)	Unclear (%)	No (%)	Not Applicable (%)
**Issues for continuous data**				
Was the choice of effect size appropriate?	11 (100%)	-	-	-
Was skew of data a potential problem that was not appropriately addressed?	-	1 (9.09%)	2 (18.18%)	8 (72.72%)
**Issues for binary data**				
Were methods appropriate to rare events/sparse data?	2 (18.18%)	-	-	9 (81.81%)
Were cut points to dichotomize continuous/ordinal outcomes justified?	3 (27.27%)	1 (9.09%)	-	7 (63.63%)
**Issues for time-to-event data**				
Were time-to-event data appropriately dealt with?	2 (18.18%)	-	-	9 (81.81%)
**Issues for ordinal data**				
Were ordinal data appropriately dealt with?	-	-	-	11 (100%)
**Indirect comparisons?**				
Were indirect comparisons performed appropriately?	2 (18.18%)	1 (9.09%)	-	8 (72.72%)

**Table 4 ijerph-20-06226-t004:** Summary judgment in parts of the tool.

Summary Judgment	Yes	Probably Yes	Unclear	Probably No	No	Not Applicable
Were the review methods adequate such that biases in location and assessment of studies were minimized or able to be identified?	11 (100%)	-	-	-	-	-
Were the individual studies analyzed appropriately and without avoidable bias?	7 (63.63%)	4 (36.36%)	-	-	-	-
Were the basic meta-analysis methods appropriate?	11 (100%)	-	-	-	-	-
Are the conclusions justified and the interpretation sound?	11/11 (100%)	-	-	-	-	-

**Table 5 ijerph-20-06226-t005:** A summary of the main results of the meta-analyses.

References	Major Results
Gulka et al. [28]	PA reduced the number of falls (RR ^c^ = 0.73, 95% CI ^d^ [0.60, 0.88]), fallers (RR = 0.80, 95% CI [0.72, 0.89]), and recurrent fallers (RR = 0.70, 95% CI [0.60, 0.81]).
Li et al. [29]	PA improved depressive symptoms among older adults compared with control groups (g ^e^ = 0.25; 95% CI [0.11, 0.38], *p* < 0.001).
Crocker et al. [30]	Physical rehabilitation improved independence in activities of daily living by 0.24 standard units (95% CI [0.11, 0.38], *p* = 0.0005) compared with control groups.
Crocker et al. [31]	Physical rehabilitation improved Barthel Index scores of six points (95% CI [2, 11], *p* = 0.008), Functional Independence Measure scores of five points (95% CI [−2, 12], *p* = 0.1), Rivermead Mobility Index scores by 0.7 points (95% CI [0.04, 1.3], *p* = 0.04), and Timed Up and Go test scores of five seconds (95% CI [−9 to 0], *p* = 0.05).
Lee and Kim [32]	PA had a preventive effect on the rate of falls (RR = 0.81, 95% CI [0.68, 0.97]).
Schoberer et al. [33]	Physical activity with a balance component (RR = 0.79, 95% CI [0.65, 0.98]) or with technical devices (RR = 0.55, 95% CI [0.30, 0.99]) reduced falls significantly. The evidence indicated that physical exercises conducted for longer than six months were beneficial (RR = 0.73, 95% CI [0.57, 0.94]).
Silva et al. [34]	PA had a preventive effect on falls (RR = 0.77, 95% CI [0.64, 0.92], *p* < 0.001). This effect was stronger when mixing several types of exercises (RR = 0.71, 95% CI [0.55, 0.90], *p* < 0.001) for at least 1–3 months (RR = 0.65, 95% CI [0.43, 0.98], *p* < 0.001) or for more than six months (RR = 0.70, 95% CI [0.56, 0.87], *p* < 0.001), with a frequency of at least 2–3 times per week (RR = 0.74, 95% CI [0.60, 0.91], *p* < 0.001).
Alvarez-Barbosa et al. [35]	Whole-body vibration improved functional mobility as assessed with the Time Up and Go test (MD = −2.49 s, 95% CI [−4.37, −0.61], *p* = 0.009).
Cao et al. [36]	PA groups showed no statistically significant differences in falls outcome (OR = 0.88, 95% CI [0.48, 1.59], *p* = 0.663) compared with control groups.
Kong et al. [37]	The Otago exercise program decreased fall risk (MD = −0.84; 95% CI [−1.17, −0.51]; *p* < 0.00001) and positively increased postural balance (MD = 5.55; 95% CI [3.60, 7.50], *p* < 0.00001), functional mobility in short distance (MD = −6.39; 95% CI [−8.07, −4.70], *p* < 0.00001), and lower-limb muscle strength (MD = 4.32; 95% CI [3.71, 4.93], *p* < 0.00001).
Wang and Tian [38]	PA significantly affected fall prevention (RR = 0.85, 95% CI [0.73, −0.98]).

RR ^c^ = risk ratio; CI ^d^ = confidence interval; g ^e^ = Hedges’s g.

## Data Availability

No new data were created.

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
