# Peer review of "Effects of Physical Activity on Physical and Mental Health of Older Adults Living in Care Settings: A Systematic Review of Meta-Analyses"

_ijerph, 2023, doi:10.3390/ijerph20136226_

Round 1
Reviewer 1 Report
Dear authors,
Thank you for doing a nice review of the meta-analyses regarding the effects of physical activity on older adults.
Please consider following issues on the revised version of this paper.
1) Title: effects of physical activity on WHAT of older adults??
2) Please add the effect size of meta-analyses in the paper and summarize them. Here, it is not obvious what are the effect sizes....
3) Please summarize the most important findings of meta-analyses in a table. You only discussed the main findings of meta-analyses in Discussion. We need to see them in results, too.
Best,
Minor English editing is needed.
Author Response
Thank you for doing a nice review of the meta-analyses regarding the effects of physical activity on older adults. Please consider following issues on the revised version of this paper.
1) Title: effects of physical activity on WHAT of older adults??
Thank you for the suggestion. We have changed the title to: Effects of Physical Activity on Physical and Mental Health of Older Adults Living in Care Settings: A Systematic Review of Meta-analyses.
2) Please add the effect size of meta-analyses in the paper and summarize them. Here, it is not obvious what are the effect sizes.... 3) Please summarize the most important findings of meta-analyses in a table. You only discussed the main findings of meta-analyses in Discussion. We need to see them in results, too.
We would like to thank the reviewer for this suggestion. Although we have provided a brief summary of the main results in section 3.3, we have elaborated our reporting based on this comment by producing Table 5, which shows the main results and the effect sizes, which are determined differently according to the meta-analysis. See Table 5.
Reviewer 2 Report
MAJOR COMMENTS
Care settings was defined at start to be an umbrella term, but then authors don’t use consistent phrasing throughout, sometimes reverting to the other phrasing (such as nursing homes or long term care homes), which is distracting or suggests that the differences might matter (they never seem to). I want to suggest that having said in Intro that you will use an umbrella term, that you try hard to be zealous in only using one phrase (except in reporting exact search phrases, I suppose), possibly with one corresponding abbreviation. The most common description in literature is probably “long term care facilities” , and “LTCFs”. If you want to use the phrase “care settings” then stick with that throughout (“CS”). Continued distinction between nursing homes or care settings or LTCHs becomes distracting, makes the reader suppose that the differences matter.
The abstract implies a synthesis but the only sythesising is narrative; please add “narrative” to explain what kind of synthesis was done, in abstract and introduction.
MINOR COMMENTS
ABSTRACT
First word of abstract… technically the word ‘several’ means 2 or 3, colloquially, it only means > 2 so in practice the word is vague & is not informative, which isn’t ideal in a scientific abstract. Rather than say “Several studies”, the words could have more useful meaning, even if the phrase used was “Many studies” or simply “Previous studies”
Please state QA tool is AMSTAR in abstract
Keywords: because this study isn’t applying meta-analysis, instead of meta-analysis as a keyword, it would be better to have a keyword about the actual methods that this study used, which is often described as a ‘meta-review’ or ‘umbrella review’. I perceive that “umbrella review” is more recognised term, and nicely doesn’t simply repeat title information.
INTRODUCTION
“ This transition in care”
Does that mean the transition between LTCF to nursing home, or a transition from community into care homes, or something else?
“a high prevalence of depressive symptoms is also observed in these people”
“these people” sounds a bit jarring, why not say “residents” ?
You could consider abbreviating physical activity to PA (throughout)
METHODS
“The full text of the eligible meta-analyses (43 items) were independently reviewed by two 102 authors (N. F. A. and F. B.) to include meta-analyses that met the following criteria”
This statement is phrased wrong. The only eligible studies are the final 11, not all 43 that were reviewed in full text.
The sentence should read something like “The studies that were not duplicates or eliminated by screening (n=43) were reviewed in full text.”
The selection diagram is mostly fine, and shows the steps correctly, although rather than “Wrong Study:” it should say “Wrong study design:”
That said, a lot could be done to make this diagram tidier and nicer looking, there is a lot of empty space
RESULTS
235 unique studies and 56,241 participants. Be careful not to imply these are net totals. Problem with this statement is that it implies a total count of unique individuals and unique original studies, which is very untrue. For instance, the participants in Becker et al. appear in meta-analyses by both Gulka et al 2020 & Lee & Kim 2016. There is also considerable overlap in Crocker 2013 primary studies.
Supplementary material, study characteristics table: please help readers by listing full bibliographic references at end of the supplementary file
DISCUSSION
Many of these points were very good, for instance Limitations does mention that the search strategy relied on category assignments being accurate in PubMED: this is very good point
I wouldn’t say that few studies are conducted on adults in these settings, far from it, it seems to be the very specific outcomes that the original MA.s were looking for that kept their included study list sometimes short
Most of the MA.s had English as language restriction, not “some”
Good points, about 1) Needing perspectives and satisfaction of adults in the LTCFs participating in these interventions; 2) stratifying results by other traits, such as cognitive ability
INTRODUCTION
According to the National Institute on Aging, nursing home is a facility
Should be
According to the National Institute on Aging, a nursing home is a facility
On the other hand, long-term care home
Should be
On the other hand, a long-term care home
METHODS
“Finally, the difference was discussed to agree on a common response.”
This is badly phrased, someone not familiar with systematic reviews won’t understand it. A better cover off statement instead could be “Any disagreements about inclusion or extraction were resolved through discussion.”
RESULTS
If they did not receive the missing data, they excluded study.
Should be rephrased
If they did not receive the missing data, they excluded that study.
DISCUSSION
However, in the present systematic review, we were unable to identify a meta-analysis on anxiety and exercise, which had been conducted with older adults in care settings. However, a recent meta-analysis…
Should be
However, in the present systematic review, we were unable to identify a meta-analysis on anxiety and exercise, which had been conducted with older adults in care settings. A recent meta-analysis….
Author Response
Care settings was defined at start to be an umbrella term, but then authors don’t use consistent phrasing throughout, sometimes reverting to the other phrasing (such as nursing homes or long term care homes), which is distracting or suggests that the differences might matter (they never seem to). I want to suggest that having said in Intro that you will use an umbrella term, that you try hard to be zealous in only using one phrase (except in reporting exact search phrases, I suppose), possibly with one corresponding abbreviation. The most common description in literature is probably “long term care facilities”, and “LTCFs”. If you want to use the phrase “care settings” then stick with that throughout (“CS”). Continued distinction between nursing homes or care settings or LTCHs becomes distracting, makes the reader suppose that the differences matter.
Thank you for the suggestion we agree that the changing terminologies can lead to confusion. We have retained the term 'care settings' and used it throughout the document.
The abstract implies a synthesis but the only sythesising is narrative; please add “narrative” to explain what kind of synthesis was done, in abstract and introduction.
Thank-you, this suggestion has been taken into account. See lines 18-19 and 78.
MINOR COMMENTS
ABSTRACT
- First word of abstract… technically the word ‘several’ means 2 or 3, colloquially, it only means > 2 so in practice the word is vague & is not informative, which isn’t ideal in a scientific abstract. Rather than say “Several studies”, the words could have more useful meaning, even if the phrase used was “Many studies” or simply “Previous studies”
Thank-you for the suggestion, 'Several studies' has been replaced by 'Previous studies'. See line 16
- Please state QA tool is AMSTAR in abstract
We thank the reviewer for this suggestion. However, the tool used to evaluate meta-analyses is not the AMSTAR tool but rather the Higgins tool, which is based on questions from the AMSTAR tool and Cochrane Handbook for Systematic Reviews of Interventions. It would not be accurate to say that the tool used is AMSTAR.
Higgins, J.P.; Lane, P.W.; Anagnostelis, B.; Anzures-Cabrera, J.; Baker, N.F.; Cappelleri, J.C.; Haughie, S.; Hollis, S.; Lewis, S.C.; Moneuse, P.; Whitehead, A. A tool to assess the quality of a meta-analysis. Res Synth Methods. 2013, 4(4):351-66. doi: 10.1002/jrsm.1092
Keywords: because this study isn’t applying meta-analysis, instead of meta-analysis as a keyword, it would be better to have a keyword about the actual methods that this study used, which is often described as a ‘meta-review’ or ‘umbrella review’. I perceive that “umbrella review” is more recognised term, and nicely doesn’t simply repeat title information.
We thank the reviewer for the suggestion, ‘Meta-analysis’ has been replaced by ‘umbrella review’ see line 28
INTRODUCTION
“ This transition in care”
Does that mean the transition between LTCF to nursing home, or a transition from community into care homes, or something else?
Thank you for drawing our attention to this point. We have therefore reworded the sentence: ‘A transition from the community into care settings’ see line 42.
“a high prevalence of depressive symptoms is also observed in these people”
“These people” sounds a bit jarring, why not say “residents”?
We thank the reviewer for the suggestion, ‘people’ has been replaced by ‘residents’ see line 51.
You could consider abbreviating physical activity to PA (throughout)
The abbreviation 'PA' was used throughout.
METHODS
“The full text of the eligible meta-analyses (43 items) were independently reviewed by two authors (N. F. A. and F. B.) to include meta-analyses that met the following criteria”
This statement is phrased wrong. The only eligible studies are the final 11, not all 43 that were reviewed in full text.
The sentence should read something like “The studies that were not duplicates or eliminated by screening (n=43) were reviewed in full text.”
Thank you for drawing our attention to this point. We have therefore reworded the sentence: The studies that were not duplicates or eliminated by screening (n=43) were reviewed in full text by two authors (N. F. A. and F. B.) to include meta-analyses that met the following criteria. See lines 101-102.
The selection diagram is mostly fine, and shows the steps correctly, although rather than “Wrong Study:” it should say “Wrong study design:”
Thank you, this correction has been made.
That said, a lot could be done to make this diagram tidier and nicer looking, there is a lot of empty space.
Thank you, we've reduced the space as much as possible.
RESULTS
235 unique studies and 56,241 participants. Be careful not to imply these are net totals. Problem with this statement is that it implies a total count of unique individuals and unique original studies, which is very untrue. For instance, the participants in Becker et al. appear in meta-analyses by both Gulka et al 2020 & Lee & Kim 2016. There is also considerable overlap in Crocker 2013 primary studies.
Supplementary material, study characteristics table: please help readers by listing full bibliographic references at end of the supplementary file.
We thank the reviewer for his suggestion. We have already proposed supplementary material in which the characteristics of the meta-analyses have been described. In order to respond to the reviewer's well-founded comment, we have subtracted all the samples from the original studies that were included in several meta-analyses. This leaves 44,477 participants, as specified in the manuscript ‘However, if we avoid repeat counting of the participants of studies that are included in several meta-analyses, we are left with 44,477 participants.’ See lines 149-150.
DISCUSSION
Many of these points were very good, for instance Limitations does mention that the search strategy relied on category assignments being accurate in PubMED: this is very good point.
I wouldn’t say that few studies are conducted on adults in these settings, far from it, it seems to be the very specific outcomes that the original MA.s were looking for that kept their included study list sometimes short
We have taken this suggestion into account by deleting this sentence from the document.
Most of the MA.s had English as language restriction, not “some”
Thank you, this correction has been addressed. See line 274
Good points, about 1) Needing perspectives and satisfaction of adults in the LTCFs participating in these interventions; 2) stratifying results by other traits, such as cognitive ability.
Comments on the Quality of English Language
All the corrections proposed by the reviewers have been considered in the final draft. We sincerely thank the reviewer for their time and their thoughtful suggestions that have improved this revised version. We have also had an English speaker review the language of the final draft to address the concerns listed below and any other language issues in the document.
INTRODUCTION
According to the National Institute on Aging, nursing home is a facility Should be According to the National Institute on Aging, a nursing home is a facility. See line 36.
On the other hand, long-term care home Should be On the other hand, a long-term care home. See line 39.
METHODS
“Finally, the difference was discussed to agree on a common response.”
This is badly phrased, someone not familiar with systematic reviews won’t understand it. A better cover off statement instead could be “Any disagreements about inclusion or extraction were resolved through discussion.” See line 140-141.
RESULTS
If they did not receive the missing data, they excluded study. Should be rephrased If they did not receive the missing data, they excluded that study. See line 196.
DISCUSSION
However, in the present systematic review, we were unable to identify a meta-analysis on anxiety and exercise, which had been conducted with older adults in care settings. However, a recent meta-analysis… Should be However, in the present systematic review, we were unable to identify a meta-analysis on anxiety and exercise, which had been conducted with older adults in care settings. A recent meta-analysis…. See line 329.